# Pixelated Instructions: Can Multimodal Large Language Models Follow Printed Instructions in Images?

## Abstract

Recent multimodal large language models (MLLMs) have shown promising instruction following capabilities on vision-language tasks. In this work, we introduce Visual Modality Instruction (VIM)[1], and investigate how well multimodal models can understand textual instructions provided in pixels, despite not being explicitly trained on such data during pretraining or fine-tuning. We adapt VIM to eight benchmarks, including OKVQA, MM-Vet, MathVista, MMMU, and probe diverse MLLMs in both the text-modality instruction (TEM) setting and VIM setting. Notably, we observe a significant performance disparity between the original TEM and VIM settings for open-source MLLMs, indicating that open-source MLLMs face greater challenges when text instruction is presented solely in image form. To address this issue, we train V-MLLM[2], a generalizable model that is capable to conduct robust instruction following in both text-modality and visual-modality instructions.

## 1 Introduction

Interleaved image-text data has been increasingly prevalent, ranging from web pages with images and tables, to user interfaces with instructions and forms, in which different modalities interact and blend together. For instance, to perform online shopping, an agent needs to understand the images, instructions and forms. Comprehensive understanding of this multi-modal data demands a range of skills, including recognizing text, understanding images, and also figuring out their interactions.

Inspired by the success of Large Language Models (LLMs), the recent research on Multimodal Large Language Models (MLLMs) Liu et al. (2023b); Dai et al. (2023); Awadalla et al. (2023); Ye et al. (2023); Zhang et al. (2023a); Su et al. (2023) may pave a way to understand this kind of vision-language data, and show promising results on a number of newly proposed benchmarks Fu et al. (2023); Yu et al. (2023); Liu et al. (2023c); Li et al. (2023); Bitton et al. (2023); Wu et al. (2023), demonstrating superb visual understanding, reasoning and generation capabilities. Despite their impressive performance, they still remain poorly understood, and also these MLLMs are not as impressive as their LLM counterparts, let alone their landing business applications. The current MLLMs are built on top of the pretrained LLMs, and *visual* instruction tuning follows the recipe from its LLM counterparts, specifically, the instruction data is synthesized by the LLMs (mostly from GPT-4 or GPT-4V) in the text format. As illustrated in the left part of Figure 1, the instruction and image are expressed in two modalities, we denote this kind of *visual* instruction data as Text-Modality Instruction (TEM). Under this setting, a pure LLM, for example, Llama 2 or Vicuna in Figure 1 can still make a plausible or correct prediction, even without accessing the image input. All the current benchmarks Fu et al. (2023); Yu et al. (2023); Liu et al. (2023c); Li et al. (2023); Bitton et al. (2023) follow the same format.

This raises a question - *how proficiently these MLLMs can follow instructions if we embed the text instruction into visual format?* As shown in the right part of Figure 1, we name it as Visual Modality Instruction, where the image and instruction are in the visual modality. Under the

---

[1] VIM is short for **VI**sual **M**odality instruction.
[2] V-MLLM is short for VIM-MLLM.

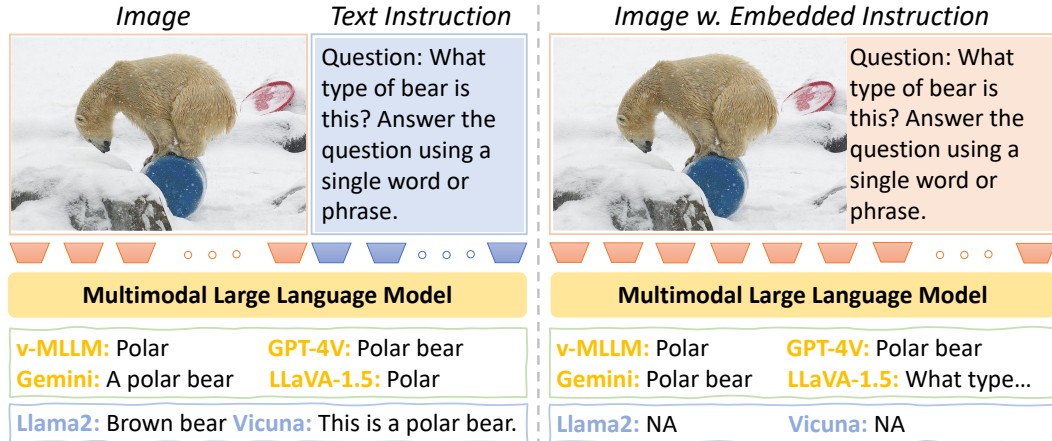

Figure 1: Evaluation paradigm comparison for MLLMs. (a) Left is TEM setting, where Image + Text instruction as two separate modalities are fed into MLLMs for inference; an LLM model (for example, Vicuna) can also make correct prediction, even without accessing to the image. (b) Right: VIM **only** takes the image modality with the text instruction embedded in the image , no additional text prompt is required, LLMs are not applicable. The above example is from OKVQA (question #209725). Note: Image modality input , Text modality input .

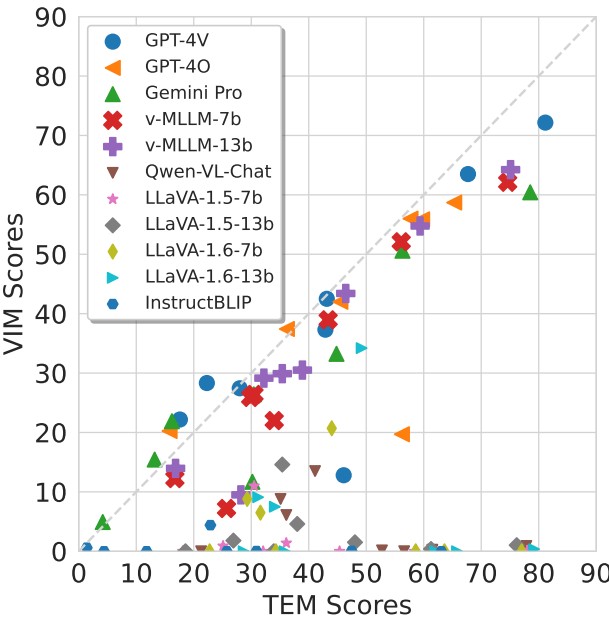

Figure 2: Model performance on the original TEM vs. VIM settings. Each data point corresponds to one model's performance on one benchmark, in total, there are 88 data points (11 models × 8 tasks). This plot reveals: 1). Open-source MLLMs (Qwen-VL-Chat, LLaVA, InstructBLIP) experience a significant performance drop from original TEM setting to VIM setting; 2). GPT-4V, GPT-4O, Gemini Pro and our V-MLLM exhibit robust instruction following capability, as their data points are consistently align closely to the diagonal line.

VIM setting, LLMs are not applicable, and LLaVA-1.5 simply repeats the question for the image, may not understand the visual-modality instruction.

Motivated by this, we introduce a new setting, called VISUAL MODALITY INSTRUCTION (short for VIM), evaluating the capability of MLLMs for visual-modality instruction following. We adapt VIM

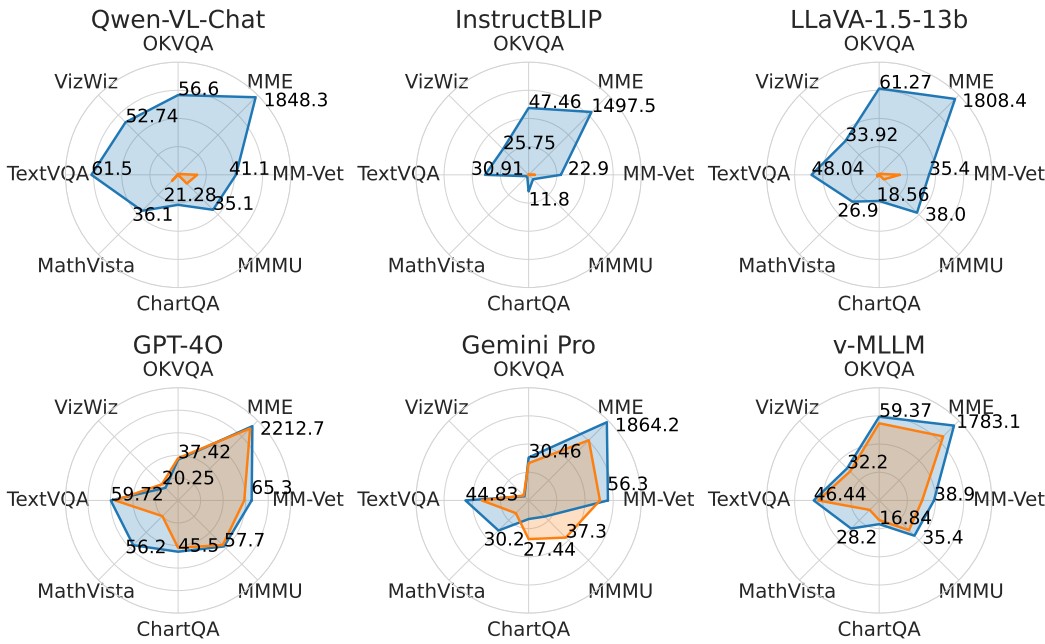

Figure 3: Performance comparison of six selected representative MLLMs for visual instruction following between text-modality instruction (TEM —) and our introduced VISUAL MODALITY INSTRUCTION (VIM —) settings on eight benchmarks. There exists a disparity between TEM and VIM settings for all open-source MLLMs (the first row); GPT-4O, Gemini Pro and our V-MLLM are robust to instruction modality.

to various benchmarks Marino et al. (2019); Fu et al. (2023); Yu et al. (2023); Lu et al. (2023); Yue et al. (2023), and compose a new benchmark - VIM-Bench. As highlighted in Figure 2 and 3, there exists a performance disparity between the TEM and VIM settings for all open-source MLLMs, all of them are not robust enough at visual-modality instruction following. To summarize, our main contributions are:

- We present VISUAL MODALITY INSTRUCTION, a challenging setting to probe the capability of Multimodal Large Language Models for visual-modality instruction following.

- We adapt the VIM to various benchmarks, and reveal a significant disparity for open-source MLLMs between their text-modality instruction setting and VIM setting.

- We train a V-MLLM, which demonstrates robust visual instruction following capabilities.

## 2 METHOD

Instruction following, is viewed as one key capability of high-performing MLLMs. In this section, we first present VIM, to examine the instruction following capability of MLLMs, specifically the visual-modality instruction following. Then, we introduce V-MLLM, enhancing the MLLMs with visual-modality instruction following.

### 2.1 VIM

#### 2.1.1 VISUAL-MODALITY INSTRUCTION

As illustrated in the left part of Figure 1, the current evaluation norm of MLLMs takes two modalities as input: image and text (as instruction). The existing MLLMs are built on top of the LLMs, benefiting from its strong text understanding capability. For the current MLLM evaluation paradigm, instruction is presented in the text modality, which can utilize the strong language priors from the LLMs for understanding. As shown in Table 3, even a pure LLM model (GPT-4, Llama 2 or Vicuna) can get

some success without accessing to the images. Interestingly, on most of eight tasks, Llama 2 shows better numbers over GPT-4 (`gpt-4-1106-preview`). We manually check some response, and find that the output from GPT-4 are more reasonable than Llama 2. This might rise several interesting issues, we leave a discussion in Section C.

VIM challenges the MLLMs by rendering the textual instruction into the visual pixel space (image), this enhancement demands not just textual but also strong visual comprehension for instruction understanding. It asks for the strong visual interpretation capability to recognize and follow the embedded instruction in the image.

### 2.1.2 DESIGN CHOICES

How to lay out the image and embedded instruction in one visual space? For zero-shot setting, there are many combinations to position the instruction and image in the same visual space. Here, we enumerate two important elements we investigated.

**Instruction Location**   Potentially, there are many options to place the instruction into the image. To narrow down the search choices of instruction's placement, we focus on three primary positions: {top, right and bottom}. Additionally, we add a random selection from these three positions to ensure robustness. Preliminary experiments[3] on these locations reveal that both GPT-4V and LLaVA-1.5 are robust to the locations of the embedded instruction, as shown in Figure 4. For the sake of simplicity, we take the bottom position as the default placement for the embedded instruction.

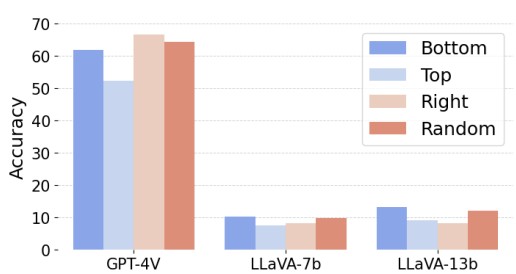

Figure 4: Exploration setup for instruction location on zero shot evaluation for MM-Vet.

**Image Resolution**   The resolution of the image is also the key for the model to understand the visual-modality instruction. For the image-text pairs, we aim to keep the resolution of the raw images, we add the text with the same font size for all images and add extra white space fillings to render the text regarding the length of the instruction. The resolution of image is minimally changed, and the origin image quality is maintained.

### 2.1.3 TEXT PROMPTING

To probe the visual instruction following of MLLMs, we prompt the models under three settings: 1) `Text-Modality Instruction` (TEM): given the image, the instruction is fed to the models as prompt in text format; 2) `Mix Instruction`: given the image with embedded instruction, an extra instruction as text prompt can be taken as input to guide models to follow the embedded instruction in the image, e.g., "*Answer the question in the image*"; 3) `Visual-Modality Instruction`: given the image with the embedded instruction, no extra text instruction is provided as text prompt, the model needs to recognize, and understand

Table 1: Exploration setup for text prompt on zero-shot evaluation for MM-Vet. * denotes from the paper reported Liu et al. (2023a).

| Models | LLM | TEM | Mix | VIM |
|---|---|---|---|---|
| *Small set* | | | | |
| GPT-4V | - | 66.7 | 54.4 | 63.5 |
| *Full Set* | | | | |
| LLaVA-1.5 | Vicuna-7b | 30.5* | 10.3 (−20.2) | 11.0 (−19.5) |
| LLaVA-1.5 | Vicuna-13b | 35.4* | 14.8 (−20.6) | 14.6 (−20.8) |
| InstructBLIP | FlanT5$_{XXL}$ | 22.9 | 12.5 (−10.4) | 4.4 (−18.5) |
| Qwen-VL-Chat | - | 41.1 | 26.2 (−14.9) | 13.5 (−27.6) |

the embedded instruction in the image automatically and follow the instruction to deliver the answer.

The `Text-Modality Instruction` setting is the standard setup in MLLMs, that the model will take the image and text question or instruction separately through vision and language encoders respectively. In the `Mix Instruction` and `Visual-Modality Instruction` settings, the models are required to understand the embedded instruction in the image, while,

---

[3]Here the small subset for preliminary experiments is 21 examples from MM-Vet.

`Mix Instruction` is a relaxed setting between `Text-Modality Instruction` and `Visual-Modality Instruction`. Preliminary results in Table 1 demonstrates that GPT-4V is robust to all three prompt settings, while, VIM is more challenging for the existing open-source MLLMs, for example, LLaVA-1.5 and Qwen-VL-Chat drop more (>20) when transferring from `Text-Modality Instruction` to the VIM setting, similarly for InstructBLIP. In this paper, we will use the VIM setting for the rest of experiments.

## 2.2 v-MLLM

### 2.2.1 VIM CORPUS

One key ingredient of high-performing MLLMs is high-quality instruction tuning data. There are two categories of visual instruction tuning data, one is the synthetic data by LLMs (i.e. GPT-4), like LLaVA Liu et al. (2023b); the other one is the synthetic data generated by GPT-4V, like LVIS-Instruct4V Wang et al. (2023) and ShareGPT-4V Chen et al. (2023). Here, we use the LVIS-Instruct4V-LLaVA-Instruct-mix880k Wang et al. (2023) as our origin instruction tuning corpus $D_R$, and convert it into the VIM format (Table 2 shows an VIM training example from GQA). We only consider the first turn for the multiple turn conversation data. In total, we get 846k VIM training data $D_V$ after filtering the unavailable image links.

### 2.2.2 VIM TRAINING

v-MLLM adopts a similar architecture with LLaVA-1.5 Liu et al. (2023a) and LVIS-Instruct4V Wang et al. (2023). The model follows an autoregressive training approach, focusing on optimizing the sequential prediction of the answer words $y_1, y_2, \ldots, y_n$ by minimizing the loss function

$$L = \sum_{i=1}^{n} \text{loss}(\text{LM}(y_{<i}, T, V), y_i)$$

where $y_{<i}$ signifies all tokens preceding the $i$-th token, $T$ and $V$ represent the textual (e.g., text-modality instruction or prompt) and visual (e.g., visual-modality instruction and image context) tokens. Here the textual token sequence $T$ is optional in the VIM training.

To train a unified model that can robustly follow the text-modality and visual-modality instructions, there are two strategies.

- `Stage-wise` training, is to train the v-MLLM on the origin corpus $D_R$ first, then continue to train the model on the VIM corpus $D_V$.
- `Mixture` training, is to combine the original corpus $D_R$ and VIM corpus $D_V$ as one corpus $D = \{D_R, R_V\}$, and train the model with in-batch random sampling.

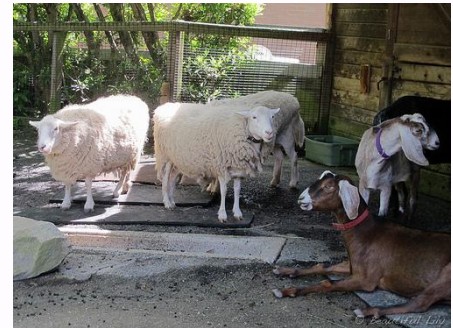

**Human:**

Question: What is the animal to the right of the sheep the collar is on? Answer the question using a single word or phrase.

**Assistant:** Goat.

Table 2: An VIM training data example from GQA.

For the `stage-wise` training, there might be one potential issue - catastrophic forgetting, the model may forget some of its behaviors after the second stage of training. Empirically, we compared the two training strategies and found there is no significant difference in Section 4.3. Following the architectures of vicuna Chiang et al. (2023) and LLaVA Liu et al. (2023b), we train two versions of the model on the top of LVIS-Instruct4V Wang et al. (2023): v-MLLM-7B and v-MLLM-13B.

## 3 EXPERIMENTS

We first build our VIM-Bench based on eight existing representative benchmarks, then compare the v-MLLM with six representative MLLMs under two settings (TEM and VIM) across all the tasks.

Table 3: Main quantitative results over each benchmark under TEM and VIM settings. ■: LLM models, ■: proprietary models, ■: the proposed models. *We use a more strict evaluation protocol to remove randomness when mapping from open-ended responses to multiple choices.

| Models | LLM | Res. | MM-Vet | MME | OKVQA | VizWiz | TextVQA | MathVista* | ChartQA | MMMU* |
|---|---|---|---|---|---|---|---|---|---|---|
| | | | | | *TEM Setting* | | | | | |
| GPT-4 | - | - | 9.8 | 74.6 | 8.37 | 2.76 | 3.36 | 18.7 | 4.12 | 28.8 |
| Llama 2 | Llama2-7b | - | 11.1 | 1609.5 | 16.21 | 5.67 | 7.18 | 23.2 | 0 | 6.2 |
| Vicuna | Vicuna-7b | - | 11.7 | 1120.6 | 4.5 | 1.88 | 1.88 | 18.1 | 0 | 2.0 |
| GPT-4V | - | - | **67.7** | 1926.6 | 22.28 | 17.59 | 43.14 | 46.1 | 28.00 | 42.9 |
| GPT-4O | - | - | 65.3 | **2212.7** | 36.20 | 15.79 | 59.72 | **56.2** | **45.50** | **57.7** |
| Gemini Pro | - | - | 56.3 | 1864.2 | 30.46 | 4.17 | 44.83 | 30.2 | 13.20 | 16.2 |
| Qwen-VL-Chat | - | - | 41.1 | 1848.3 | 56.6 | **52.74** | 61.5 | 36.1 | 21.28 | 35.1 |
| InstructBLIP | FlanT5$_{XXL}$ | 224 | 22.9 | 1497.5 | 47.46 | 25.75 | 30.91 | 1.4 | 11.80 | 4.40 |
| LLaVA-1.5 | Vicuna-7b | 336 | 30.5 | 1851.5 | 58.41 | 32.08 | 45.36 | 25.1 | 18.08 | 36.1 |
| LLaVA-1.5 | Vicuna-13b | 336 | 35.4 | 1808.4 | 61.27 | 33.92 | 48.04 | 26.9 | 18.56 | 38.0 |
| LLaVA-1.6 | Vicuna-7b | | 44 | 1828.6 | 58.6 | 34.29 | 63.61 | 31.6 | 22.76 | 29.3 |
| LLaVA-1.6 | Vicuna-13b | | 49.2 | 1880.5 | **62.01** | 35.81 | **65.79** | 34.1 | 26.68 | 31.2 |
| v-MLLM | Vicuna-7b | 336 | 29.9 | 1771.1 | 56.09 | 30.48 | 43.38 | 25.7 | 16.72 | 34.0 |
| v-MLLM | Vicuna-13b | 336 | 38.9 | 1783.1 | 59.37 | 32.20 | 46.44 | 28.2 | 16.84 | 35.4 |
| | | | | | *VIM Setting* | | | | | |
| GPT-4V | - | - | **63.5** | 1713.1 | 28.32 | 22.18 | 42.50 | 12.8 | 27.44 | 37.3 |
| GPT-4O | - | - | 58.7 | **2144.3** | 37.42 | 20.25 | 55.88 | **19.7** | **42.00** | **56.0** |
| Gemini Pro | - | - | 50.6 | 1434.6 | 26.43 | 4.93 | 33.24 | 11.7 | 15.44 | 21.9 |
| Qwen-VL-Chat | - | - | 13.5 | 21.2 | 0.01 | 0.15 | 0.27 | 6.1 | 0 | 8.8 |
| InstructBLIP | FlanT5$_{XXL}$ | 224 | 4.40 | 0 | 0.07 | 0 | 0.04 | 0.6 | 0 | 0 |
| LLaVA-1.5 | Vicuna-7b | 336 | 11.0 | 2.9 | 0 | 0 | 0 | 0.9 | 0 | 1.4 |
| LLaVA-1.5 | Vicuna-13b | 336 | 14.6 | 24.4 | 0.38 | 0 | 1.51 | 1.8 | 0 | 4.6 |
| LLaVA-1.6 | Vicuna-7b | | 20.7 | 0 | 0 | 0 | 0 | 6.5 | 0 | 8.8 |
| LLaVA-1.6 | Vicuna-13b | | 34.2 | 8.33 | 0 | 0 | 0.01 | 7.5 | 0 | 9.1 |
| v-MLLM | Vicuna-7b | 336 | 25.9 | 1474.6 | 52.10 | 26.40 | 38.96 | 7.2 | 12.24 | 22.0 |
| v-MLLM | Vicuna-13b | 336 | 30.5 | 1525.1 | **54.76** | **29.15** | 43.40 | 9.5 | 13.96 | 29.9 |

## 3.1 VIM-BENCH

**Benchmarks**  To assess the generalization capability of MLLMs, we adapt VIM to eight representative benchmarks, including MME Fu et al. (2023), MM-Vet Yu et al. (2023), OKVQA Marino et al. (2019), VizWiz Bigham et al. (2010), TextVQA Singh et al. (2019), MathVista Lu et al. (2023), ChartQA Masry et al. (2022), and MMMU Yue et al. (2023). The details of source datasets, data processing pipeline, and evaluations can be found in Appendix A.

**Data Reformatting**  Given the above mentioned benchmarks, we try to do minimal changes (i.e., keeping the image resolution) for evaluation. This process involves reformatting text instruction into visual-modality instruction by moving the text instruction into the image modality. In reformatting, we retain the original task's goal while maintaining the original images with text instructions rendering at the bottom of the image, see the example in Figure 1. These repurposed benchmarks are integrated into our VIM-Bench. Theoretically, VIM can be applied to any existing benchmarks, even for pure NLP tasks. We choose eight representative MLLM benchmarks, although our selections are not exhaustive, they provide a broad basis for MLLM evaluation.

## 3.2 EVALUATION SETUP

**Evaluation Settings**  For each benchmark, we have two main evaluation settings, one is the standard evaluation setting, which is Text-Modality Instruction, `image + text prompt` as the input, denoted as TEM setting; the other one is the VIM setting with only `image` as input. For the VIM setting, it also can accept any text prompts, we call `Mix Instruction` in the ablation experiments.

**Evaluation Metrics**  For the evaluation metrics, we follow the evaluation protocols and metrics of each benchmark. For MathVista and MMMU, the open-ended responses will be parsed into options before calculating the accuracy. We take a more strict protocol that requires the models to follow

Table 4: MLLMs' instruction recognition response to the question #3575865 in OKVQA.

<table>
<tr><td>Image w. Embedded Instruction</td><td>Text Prompt: What is the text in the image?</td></tr>
<tr><td rowspan="2">Question: What toy is this? Answer the question using a single word or phrase.</td><td>Recognized Instructions</td></tr>
<tr><td>LLaVA-1.5-13B: The image shows a man sitting in a pew with a teddy bear on his back. The teddy bear is wearing a backpack, and the man appears to be looking at it. The scene takes place in a church, with several other people present in the background.

GPT-4V: The text in the image says: "Question: What toy is this? Answer the question using a single word or phrase."</td></tr>
</table>

explicit instructions (e.g., "Answer with the choice letter."), instead of mapping to the closest options no matter whether the response is related or not. Details of metrics are illustrated in Appendix A.3.

## 3.3 BASELINES

We evaluate nine representative open-source and proprietary MLLMs (with different variations), including InstructBLIP Dai et al. (2023), LLaVA-1.5 Liu et al. (2023a), LLaVA-1.6 Liu et al. (2024), Qwen-VL-Chat Bai et al. (2023), GPT-4V OpenAI (2023a), Gemini Pro Team et al. (2023), and latest GPT-4O OpenAI (2024). For LLaVA-1.5, we use its latest versions with Vicuna-v1.5 as the LLM backbone. We use Flan-T5 XXL Chung et al. (2022) as LLM backbone for InstructBLIP. All experiments are conducted on NVIDIA A100 and H100 80G GPUs, we use as large models as possible, model training was integrated with Deepspeed Zero-3. For 7b size models, we use batch size 80 with 3 epochs; for 13b models, batch size is 48 with 2 epochs on a single node with 8 GPUs.

## 3.4 MAIN RESULTS

Table 3 summarizes the overall results for two settings. 1). In the original TEM setting, the backbone LLM models can get decent performance on these benchmarks, even without access to the image modality. Interestingly, on six out of eight tasks, Llama 2 is much better than GPT-4, we will briefly discuss this issue in Section C. 2). For all open-source MLLMs, there is a significant performance disparity between the TEM setting and VIM setting. 3). GPT-4V and Gemini Pro are robust to the instruction modality, while, open-source MLLMs struggle in the VIM setting, achieve significantly low scores. 4). Our proposed V-MLLM shows robust instruction following capabilities in two settings across all the tasks, especially in the VIM setting, significant gain over open-source MLLMs.

## 4 ABLATION

We have demonstrated that VIM is a challenging setting for the current open-source MLLMs. To better understand the gap between the standard `Text-Modality Instruction` and `Visual-Modality Instruction`, we decompose the VIM into two steps for ablation.

## 4.1 INSTRUCTION RECOGNITION

One hypothesis is the vision encoder of MLLMs cannot discern the instruction in the image, only generates the long description for the image, as the LLaVA example shown in Table 4. *Can the MLLMs recognize the embedded instruction in the image?* Here, we conduct an experiment to verify the visual-modality instruction recognition ability of the models, we choose an external dataset VQA$_{v2}$ for test (beyond our eight test benchmarks)[4], we explicitly prompt the models with "*What is*

---

[4]Due to the large size of VQA$_{v2}$, it is expensive to run the full *test-dev* and *test* splits for GPT-4V and Gemini Pro for fair comparison, we did not include it in the VIM-Bench.

Table 5: Ablation results on OKVQA, MM-Vet, VizWiz tasks.

| Models | LLM | OKVQA | | MM-Vet | | VizWiz | |
|---|---|---|---|---|---|---|---|
| | | VIM | MIX | VIM | MIX | VIM | MIX |
| LLaVA-1.5 | Vicuna-7b | 0.00 | 14.28 (+14.28) | 11.0 | 10.3 (-0.7) | 0 | 22.47 (+22.47) |
| InstructBLIP | FlanT5$_{XXL}$ | 0.07 | 25.44 (+25.37) | 4.4 | 12.5 (+8.1) | 0 | 18.79 (+18.79) |
| Qwen-VL-Chat | - | 0.01 | 30.75 (+30.74) | 13.5 | 26.2 (+12.7) | 0.15 | 34.03 (+33.88) |
| GPT-4V | - | 28.32 | 27.70 (-0.62) | 63.5 | 54.4 (-9.1) | 22.18 | 23.37 (+1.19) |
| v-MLLM | Vicuna-7b | 52.10 | 51.82 (-0.28) | 25.9 | 24.4 (-1.5) | 26.40 | 27.34 (+0.94) |

*the text in the image?*", then manually check the outputs with the ground-truth text instructions in the zero-shot setting for VQA$_{v2}$. We select 30 images as our subsets for both models. Here, the images we choose only have the embedded instructions since we want to verify the instruction recognition capability of these models in an ideal setup.

We do `word match` and `semantic match` for the results. For example, the origin instruction of question #393225000 in VQA$_{v2}$ is "*What website copyrighted the picture*", the output "*What website copied the picture?*" is counted as a correct word match, since most of the words are recognized. However, it will be considered as a wrong semantic match. As shown in

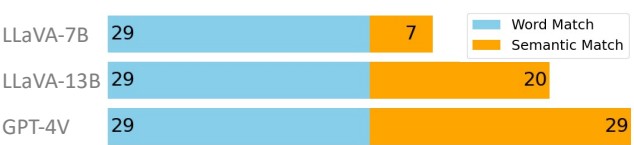

Figure 5: Instruction recognition results on VQA$_{v2}$. We report the correct number of matches out of total 30 samples for LLaVA-1.5-7B & 13B, and GPT-4V.

Figure 5, GPT-4V can recognize the embedded instructions nearly perfectly on word match and semantic match, the only "*failure*" case is from an image containing a text logo inside as shown in Table 10 (in the Appendix). While, LLaVA-1.5 can detect words of the instructions, but semantically different. Table 6 showcases some example results for zero shot instruction recognition. GPT-4V can recognize the text instruction in both settings, LLaVA can detect some words of the instructions, but may not perfectly recognize the instructions. A more detailed analysis and examples of instruction recognition capability can be found in Appendix B.2.

## 4.2 INSTRUCTION FOLLOWING

We further relax the VIM setting to the `Mix Instruction` setting with an extra text prompt "*Answer the question in the image in one word or phrase.*", these open-source MLLMs exhibit some degree of success (the highlighted green columns in Table 5), which proves that the existing MLLMs rely more on their LLM backbones for instruction following. For example, on OKVQA and VizWiz, LLaVA-1.5, InstructBLIP and Qwen-VL-Chat can achieve moderate success (+10) in the `Mix Instruction` setting, significant improvement over their VIM setting.

## 4.3 MIXTURE TRAINING V.S. STAGE-WISE TRAINING

We ablation the two training strategies on the whole corpus, and verify the results on three downstream tasks under both the TEM and VIM settings. Table 7 shows that 1). Training procedure is often unstable, hard to seek a well balanced checkpoint on all the tasks, which is consistent with the observation in screenshot LM Gao et al. (2024). 2). There is no significant difference for mixture training and stage-wise training, we use stage-wise training in our experiments. 3). Referring to LLaVA, v-MLLM maintains comparable performance under the TEM setting and achieves significantly better performance under the VIM setting.

Table 6: Zero Shot Instruction Recognition: MLLMs's recognition to the example questions in VQA.

*Image w. Embedded Instruction*   #42000 in VQA

Question: What color are the gym shoes? Answer the question using a single word or phrase.

**Text Prompt:** What is the text in the image?

**GPT-4V:** Question: What color are they gym shoes? Answer the question using a single word or phrase.

**LLaVA-1.5-7B:** The text in the image is a question asking, "What color are the gym shoes?"
**LLaVA-1.5-13B:** The text in the image is a question asking about the color of the gym shoes.

*Image w. Embedded Instruction*   #757000 in VQA

Question: How many elephants are pictured in this photo? Answer the question using a single word or phrase.

**Text Prompt:** What is the text in the image?

**GPT-4V:** Question: How many elephants are pictured in this photo? Answer the question using a single word or phrase.

**LLaVA-1.5-7B:** The text in the image is a question asking how many elephants are pictured in the photo.
**LLaVA-1.5-13B:** The text in the image is a question asking how many elephants are pictured in the photo.

Table 7: Ablation results on training strategies. Note: Taking LLaVA-1.5 as referring baseline.

| Models | LLM | Training | OKVQA | | MM-Vet | | TextVQA | |
|---|---|---|---|---|---|---|---|---|
| | | | TEM | VIM | TEM | VIM | TEM | VIM |
| LLaVA-1.5 | Vicuna-7b | - | 58.41 | 0 | 30.5 | 11.0 | 45.36 | 0 |
| v-MLLM | Vicuna-7b | Stage-wise | 56.09 | 52.10 | 29.9 | 25.9 | 43.38 | 38.96 |
| v-MLLM | Vicuna-7b | Mixture | 58.74 (+2.65) | 52.90 (+0.8) | 28.8 (-1.1) | 23.5 (-2.4) | 45.49 (+2.11) | 41.77 (+2.81) |
| LLaVA-1.5 | Vicuna-13b | - | 61.27 | 0.38 | 35.4 | 14.6 | 48.04 | 1.51 |
| v-MLLM | Vicuna-13b | Stage-wise | 59.37 | 54.76 | 38.9 | 30.5 | 46.44 | 43.40 |
| v-MLLM | Vicuna-13b | Mixture | 57.84 (-1.53) | 53.09 (-1.67) | 37.6 (-1.2) | 28.6 (-1.9) | 45.77 (-0.67) | 43.42 (+0.02) |

### 4.4 QUALITATIVE OBSERVATIONS

**People related questions**   In VQA and MM-Vet, there are some categories of questions about people or movies. For example, in Table 8, GPT-4V will response "*Sorry, I cannot help with that.*", while, LLaVA-1.5 and InstructBLIP just simply repeat the question. We highlight several intriguing observations during the experiments, and a comprehensive qualitative analysis is provided in the Appendix B.3.

## 5 RELATED WORK

**Multimodal Large Language Models**   With the success of Large Language Models (LLMs) OpenAI (2023a); Google (2023); Touvron et al. (2023a;b), there is growing interest in studying Multimodal Large Language Models Liu et al. (2023b); Dai et al. (2023); Zhu et al. (2023); Awadalla et al. (2023); Ye et al. (2023); Zhang et al. (2023a); Su et al. (2023); Sun et al. (2023); Bai et al. (2023) to improve multimodal understanding, reasoning and generation by leveraging the strong capability of LLMs. Following the recipe of the instruction tuning Taori et al. (2023); Chiang et al. (2023); Peng et al. (2023) in LLMs, LLaVA Liu et al. (2023b) and MiniGPT-4 Zhu et al. (2023) propose

Table 8: MLLMs's response to the question #202 in MM-Vet.

| *Image w. Embedded Instruction* | **Models' Responses** |
|---|---|
| 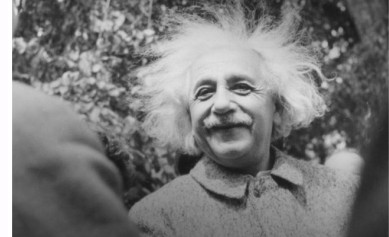  Question: Can you give a short introduction to this person? | **GPT-4V:** Sorry, I cannot help with that.  **LLaVA-1.5:** Question: Can you give a short introduction to this person?  **InstructBLIP:** albert einstein with the caption can you give a short introduction to this person?  **v-MLLM:** albert einstein |

to use GPT-4 OpenAI (2023a) to synthesize the instruction, and employ the open-source LLMs for instruction tuning to connect the pretrained vision encoder and open-source LLMs. Recently, GPT-4V OpenAI (2023b) and Gemini Team et al. (2023) were released, followed by work Yang et al. (2023); Zhang et al. (2023b); Tong et al. (2024) that explore their impressive multimodal capabilities. LVIS-Instruct4V Wang et al. (2023) and ShareGPT-4V Chen et al. (2023) are created with GPT-4V for high-quality instruction tuning data. Our work implies the limitations of existing open-source MLLMs, and provides a solution to enhance robust visual instruction following.

There is another thread of work Rust et al. (2022); Lee et al. (2023); Gao et al. (2024) emerged to explore the text along with images, charts, and tables all through visual input. Among of them, the most similar with ours is Pix2Struct Lee et al. (2023), however, there are a few difference: 1). Data-wisely, Pix2Struct takes the web-scale interleave data (HTML Dom Tree) for pretraining, we utilize the existing public image-text corpus (mainly GPT-4V synthetic corpus) for VIM training. 2). Model-wisely, Pix2Struct employs an encoder-decoder model with BART-like learning signals, v-MLLM is an image encoder-decoder model (based on the pretrained LLM backbone) with autoregressive loss. 3). For downstream tasks, Pix2Struct does the finetuning before evaluation. While, v-MLLM training has no individual downstream finetuning.

**Benchmarks** In parallel with the MLLMs development, a trend emerges in proposing a variety of benchmarks. MME Fu et al. (2023) proposes 14 tasks with Yes/No questions based on the images. MMBench Liu et al. (2023c) and SEED-Bench Li et al. (2023) introduce benchmarks that cover a variety of multiple-choice questions. MM-Vet Yu et al. (2023) extends to evaluate the open-ended outputs from MLLMs. VisIT-Bench Bitton et al. (2023) accesses a range of tasks from recognition to complex reasoning, while Q-Bench Wu et al. (2023) accesses low-level visual perception and understanding. MathVista (Lu et al., 2023) focuses on systematically studying the mathematical reasoning capability in visual context. More recently, MMMU (Yue et al., 2023) is proposed to evaluate multimodal models in a broad range of college-level subject knowledge. All these benchmarks focus on text instruction evaluation for MLLMs, and our proposed VIM integrates the text instruction into the visual modality space, asking for strong visual interpretation capability for embedded instruction recognition and following. Additionally, our VIM is orthogonal with these existing benchmarks, can be seamlessly adapted to any of them.

## 6 CONCLUSION

In this work, we review the existing MLLMs from a visual perspective, and present VIM, a challenging setting to assess the visual instruction following ability of Multimodal Large Language Models. We adapt VIM to eight benchmarks, leading to VIM-Bench. Through in-depth probing under zero-shot setting, we observed a common issue for the existing open-source MLLMs: all fall short in the VIM setting, in most cases performing not as good as those in the original TEM setting. Furthermore, we train v-MLLM, which demonstrates robust instruction following capabilities under text and visual modality instruction settings on all the tasks.

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

# Part I

# Appendix

## A BENCHMARK DETAILS

### A.1 SOURCE DATASETS

We consider eight representative datasets, MM-Vet Yu et al. (2023), MME Fu et al. (2023), OKVQA, VizWiz, TextVQA, MathVista, ChartQA, and MMMU. We also provide probing analysis on $VQA_{v2}$ *test-dev* split, RefCOCO *testA* split, RefCOCO+ *testA* split, and RefCOCOg *test* split to better illustrate how the data is formatted.

In Figure 6, we showcase an example from the source dataset $VQA_{v2}$, which is with instruction probing setting "Text". We also consider other two probing settings, "MIX" that have an additional text prompt to guide the visual embedded instruction following, and our proposed "VIM" that only allow the image with embedded instruction as input. We also showcases dataset examples sampled from source datasets RefCOCO, MME, and MM-Vet. All the VIM test samples does not include any additional instruction input in text modality (noted as "NA").

### A.2 DATASET PROCESSING PIPELINE

For each source dataset, we start by building up zero shot by embedding instructions into the input image to concatenate as a new image which contains instructions in image modality. In this way, we obtain a new image with embedded instructions for each image-question pair.

### A.3 EVALUATION DETAILS

**Metrics**  We follow the evaluation pipeline for each benchmark. We use parsing and accuracy for MathVista and MMMU with a more strict protocol. For example, when the model is outputting an empty or random string for a multiple-choice question, in MathVista, the original evaluation protocol will use Levenshtein distance to map to a most similar prediction option, and in MMMU, a random choice from the candidate list will be applied. This will introduce noise and randomness for the evaluation, may not correctly reflect the model performance. In our strict evaluation protocol, we eliminate this random match strategy.

For OKVQA and TextVQA, we follow the leaderboard evaluation to use an evaluation metric that is robust to inter-human variability: $\text{Acc(ans)} = \min\left\{\frac{\#\text{humans that said ans}}{3}, 1\right\}$. For ChartQA, we use relaxed accuracy on human and augmented split.

For MME, the standard metric ($Score$) proposed in Fu et al. (2023) is the summed up Accuracy ($Acc$) and Accuracy+ ($Acc_+$) as: $Score = sum(Acc \times 100\%, Acc_+ \times 100\%)$, where the former one count each correct answer as correct, while the latter one only considers correct when both "Yes" and "No" questions for each image are answered correctly.

For MM-Vet, we use GPT-4 ("gpt-4-0613" version) to automatically provide the score for each sample. The final Accuracy reported as: $\text{Acc} = \frac{\sum_{i=1}^{N} s_i}{N} \times 100\%$, where $s_i$ is the score at scale $0 - 1$ for sample $i$.

## B FULL ANALYSIS

### B.1 ROBUSTNESS ANALYSIS

#### B.1.1 PROMPT FOR MIX PROBING SETTING

In `mix` probing setting, the MLLMs can accept an extra text instruction input as guidance. The model performance may vary when given different prompts. We report the results using four relevant but

Table 9: Zero shot evaluation results of `Text` probing setting on VQA$_{v2}$, MME, MM-vet. This is the popular setting for evaluating text instruction following capability of MLLMs, where the input image and text instruction are both provided.

| Models | LLM | Embedded Instruction | Zero shot | | |
|---|---|---|---|---|---|
| | | | VQA$_{v2}$ | MME | MM-Vet |
| | | Sub set | | | |
| LLaVA-1.5 | Vicuna-7b | w.o. | 60.75 | 108 | 31.3 |
| | | w. | 57.88 | 88 | 27.9 |
| | Vicuna-13b | w.o. | 61.00 | 106 | 35.2 |
| | | w. | 58.00 | 87 | 32.7 |

diversified prompts (Prompt #1-#5) in Table 13. To be specific, the detailed prompts we use are: 1) Prompt #1: *"Answer the question in the image."*, 2) Prompt #2: *"Please answer the question that is written in the image."*, 3) Prompt #3: *"Follow the instruction embedded in the image."*, 4) Prompt #4: *"Detect the question in the image and directly answer to it."*.

### B.1.2 IMAGE EMBEDDED WITH INSTRUCTION

To investigate whether the model performance is robust to the minimal changes introduced by the embedded instruction, we give both the original instruction in the text modality and the image with instruction embedded as the image modality to the model.

In Table 9, we present comparative results of LLaVA-1.5 using Vicuna-7b and Vicuna-13b language backbones. It's observed that embedding images with instructions leads to a marginal decline in performance. This trend suggests that current MLLMs may not be entirely robust to variations in images. However, this performance degradation is minor and within acceptable limits. This implies that the disparity in performance between the `Text` and VIM probing settings is not solely attributable to changes in the images, but is largely due to the models' capacity to follow visual embedded instructions.

### B.2 DETAILS OF INSTRUCTION RECOGNITION

Based on Section 4.1, we conduct an ablation to verify the instruction recognition capability of these MLLMs. Table 15 showcases some example results for zero shot instruction recognition. GPT-4V can recognize the text instruction in both settings, LLaVA can detect some words of the instructions, but may not perfectly recognize the instructions, especially in the one shot setting. Table 10 shows a failure example of GPT-4V for Instruction Recognition, it recognizes the logo texts on the bus as the text instruction.

### B.3 QUALITATIVE OBSERVATIONS

### B.3.1 GROUNDING OF GPT-4V

Table 11 shows that GPT-4V's grounding capability can be unleashed when carefully prompted.

## C LIMITATIONS

We discuss the limitations of our work as follows: 1). Though V-MLLM exhibits robust instruction following capability in both the TEM and VIM settings, it still has a gap with proprietary models, especially GPT-4V, there is still space to improve to be a generalist model. 2). In this work, the evaluation protocols and metrics of VIM setting follow these from the original TEM setting, it also inherits the shortcoming of these evaluations, we leave these for future work. 3). For proprietary

Table 10: One "failure" examples to show the instruction recognition of GPT-4V.

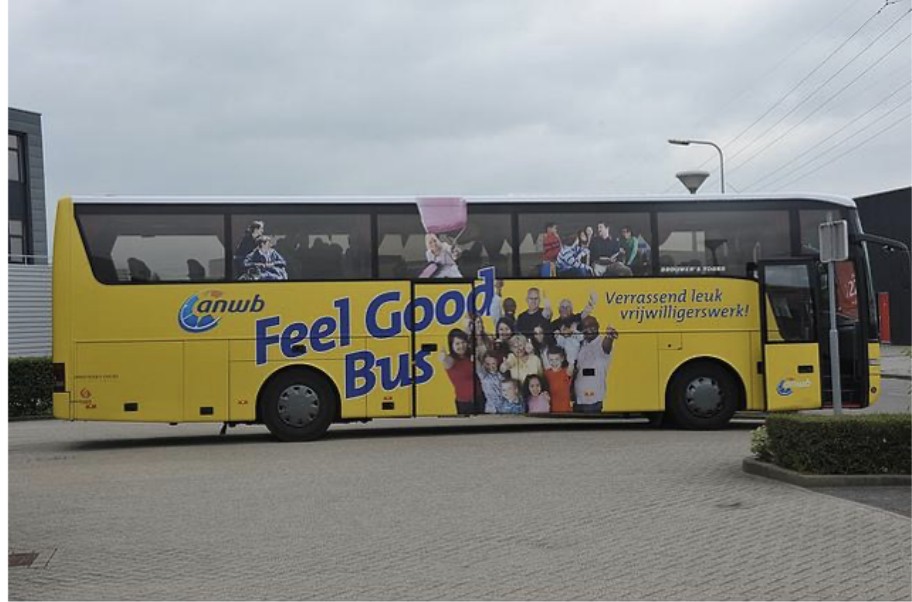

*Image w. Embedded Instruction*

Question: How many girls are on the advertisement?
Answer the question using a single word or phrase.

**Text Prompt :**
What is the text in the image?

**GPT-4V :**
1. "anwb"
2. "Feel Good Bus"
3. "Verrassend leuk vrijwilligerswerk!"

models (GPT-4V and Gemini Pro), they are evolving with in-context learning as more queries are fed to the models, the results from these models may not be stable or potentially reproducible.

**MLLM Evaluation**   In the LLM exploration setting, Table 3 shows that Llama 2 is much better than GPT-4 on six of eight multimodal tasks, without image input. We manually check some results, and find that the responses from GPT-4 are more reasonable than Vicuna and Llama 2. For example, in Table 14, we only provide the text `Prompt` to the LLMs, the output from Vicuna and Llama 2 is more like text continuation based on the training corpus, while, GPT-4 makes more reasonable response to the `Prompt` question.

Another potential issue exposed from this exploration experiment is about MLLM evaluation, though GPT-4 makes more reasonable response, its score is low on all six tasks, which may be contrary to the objective of these benchmarks. In-depth analysis and discussion of evaluation protocol and metrics are beyond the scope of this work, which may leave for future work.

## D   BROADER IMPACTS

**Evaluation and Safety**   Safety is a critical aspect of MLLMs, particularly when these models are used in the real-world applications. The safety of MLLMs is assessed through various benchmarks that test their robustness against unsafe instructions and harmful content. VIM identified a common

Table 11: Carefully curated prompting to unleash the grounding capability of GPT-4V.

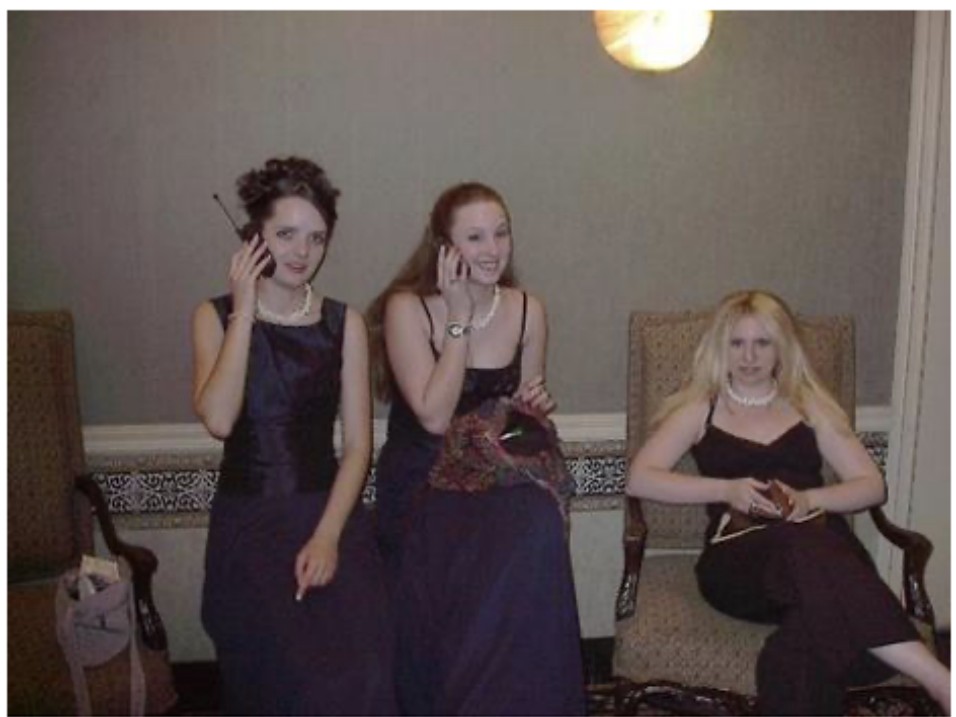

*Image w.o. Embedded Instruction*

**Text Prompt :**
How many people in the image? Give them a label and show me their normalized coordinate in the format [x1, y1, x2, y2] rounded to four decimal places.

**GPT-4V :**
There are 3 people in the image.
Labels and normalized coordinates for the people are:
1. Person A: [0.1081, 0.0432, 0.3784, 0.7973]
2. Person B: [0.3838, 0.0000, 0.7027, 0.8108]
3. Person C: [0.7027, 0.2973, 0.9676, 0.8108]

Table 12: Zero shot evaluation results on MM-Vet with detail scores across six core vision and language capabilities.   highlights the numbers for mix instruction setting. The numbers are presented in %. Rec: Recognition, Know: Knowledge, Gen: Language Generation, Spat: Spatial Awareness.

|              | Rec  | OCR  | Know | Gen  | Spat | Math | Total |
|--------------|------|------|------|------|------|------|-------|
| LLaVA-1.5-7b | 12   | 6.6  | 5.6  | 5.6  | 6.4  | 3.8  | 10.1  |
|              | 9.7  | 7.7  | 5.1  | 3.1  | 6.9  | 3.8  | 8.5   |
| LLaVA-1.5-13b| 15.2 | 13.6 | 6.9  | 10.9 | 11.9 | 3.8  | 14.4  |
|              | 18.5 | 15.7 | 7.3  | 9.5  | 15.3 | 9.6  | 16.9  |
| InstructBLIP | 6.4  | 1.6  | 1.8  | 1.2  | 2.7  | 0    | 4.4   |
|              | 14.5 | 7.8  | 2.6  | 0.9  | 9.3  | 11.5 | 12.5  |
| GPT-4V       | 61.4 | 65.2 | 51.2 | 53.7 | 67.6 | 59.2 | 63.5  |

issue for the existing open-source MLLMs, it may help to improve the robustness of the MLLMs; and also improve the current evaluation benchmarks of MLLMs.

Table 13: Zero shot evaluation results on MME subset under `mix` setting. We compare the performance of LLaVA-1.5-7b and LLaVA-1.5-13b across four different prompts.

| Task | LLaVA-1.5-7b | | | | LLaVA-1.5-13b | | | |
|---|---|---|---|---|---|---|---|---|
| | Prompt #1 | Prompt #2 | Prompt #3 | Prompt #4 | Prompt #1 | Prompt #2 | Prompt #3 | Prompt #4 |
| artwork | 5 | 5 | 0 | 0 | 0 | 0 | 0 | 0 |
| celebrity | 5 | 5 | 0 | 0 | 1 | 0 | 0 | 0 |
| code reasoning | 4 | 3 | 0 | 0 | 0 | 0 | 0 | 0 |
| color | 5 | 5 | 0 | 0 | 1 | 1 | 0 | 0 |
| commonsense reasoning | 4 | 4 | 0 | 0 | 0 | 0 | 0 | 0 |
| count | 5 | 5 | 0 | 0 | 1 | 0 | 0 | 0 |
| existence | 6 | 6 | 0 | 0 | 0 | 0 | 0 | 0 |
| landmark | 5 | 5 | 0 | 0 | 3 | 0 | 0 | 0 |
| numerical calculation | 5 | 5 | 0 | 0 | 0 | 0 | 0 | 0 |
| OCR | 5 | 5 | 0 | 0 | 0 | 0 | 0 | 0 |
| position | 4 | 5 | 0 | 0 | 0 | 0 | 0 | 0 |
| posters | 5 | 5 | 0 | 0 | 1 | 0 | 0 | 0 |
| scene | 5 | 5 | 0 | 0 | 0 | 0 | 0 | 0 |
| text translation | 5 | 5 | 0 | 0 | 0 | 0 | 0 | 0 |
| *Correct* | 68 | 68 | 0 | 0 | 7 | 1 | 0 | 0 |

Table 14: LLMs' responses to the questions in MM-Vet and OKVQA. (*Image* is not feed into the LLMs.)

**MM-Vet #1**

*Image*:

Solve the following equations:

1) $8x + 11 = 4x + 14$

2) $7d - 4 = 11d - 9$

*Text Prompt*: What is d in the last equation? Answer the question in one word or phrase.

**GPT-4:** I'm sorry, but I need more context or information about the "last equation" you are referring to in order to provide an answer. Could you please provide the equation or more details surrounding it?
**Vicuna:** d is the derivative of the function with respect to x.
**Llama 2:** D is the constant of integration.

**OKVQA #2971475**

*Image*:

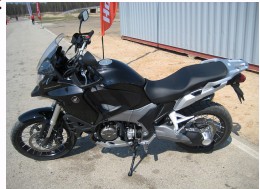

*Text Prompt*: What sport can you use this for? Answer the question in one word or phrase.

**GPT-4:** Please provide the context or the object you're referring to in order to answer the question about which sport it can be used for.
**Vicuna:** Basketball.
**Llama 2:** Basketball

**Ethical Considerations**  The deployment of MLLMs necessitates careful consideration of ethical implications, including privacy, bias, and the potential misuse of technology. Ensuring that these models are properly developed and used responsibly is crucial to mitigate risks and maximize their positive impact on society.

**Practical Applications**  MLLMs have a wide range of applications, from enhancing accessibility in technology to improving human-computer interactions. There are many real application scenarios for VIM, like UI interface navigation, agent development etc. This can improve efficiency, accuracy, and safety in manufacturing, logistics, and other sectors where visual instruction is crucial. We hope the VIM can benefit for the MLLM's evolution.

Table 15: Zero Shot Instruct Recognition: MLLMs's recognition to the example questions in VQA.

*Image w. Embedded Instruction* #42000 in VQA

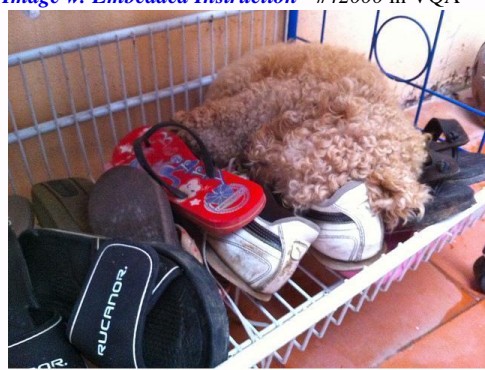

Question: What color are the gym shoes? Answer the question using a single word or phrase.

**Text Prompt:** What is the text in the image?

**GPT-4V:** Question: What color are they gym shoes? Answer the question using a single word or phrase.

**LLaVA-1.5-7B:** The text in the image is a question asking, "What color are the gym shoes?"
**LLaVA-1.5-13B:** The text in the image is a question asking about the color of the gym shoes.

*Image w. Embedded Instruction* #757000 in VQA

Question: How many elephants are pictured in this photo? Answer the question using a single word or phrase.

**Text Prompt:** What is the text in the image?

**GPT-4V:** Question: How many elephants are pictured in this photo? Answer the question using a single word or phrase.
**LLaVA-1.5-7B:** The text in the image is a question asking how many elephants are pictured in the photo.
**LLaVA-1.5-13B:** The text in the image is a question asking how many elephants are pictured in the photo.

*Image w. Embedded Instruction* #132132000 in VQA

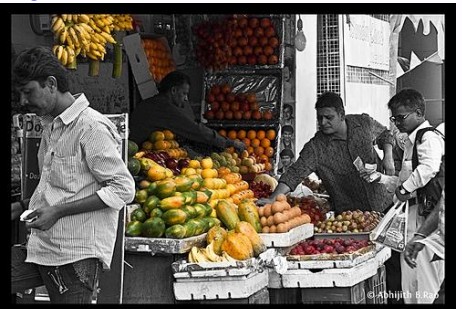

Question: How many men are shopping? Answer the question using a single word or phrase.

**Text Prompt:** What is the text in the image?

**GPT-4V:** Question: How many men are shopping? Answer the question using a single word or phrase.

**LLaVA-1.5-7B:** The text in the image is a question asking, "How many men are shopping?"
**LLaVA-1.5-13B:** The text in the image is a question asking how many men are shopping.

*Image w. Embedded Instruction* #131612000 in VQA

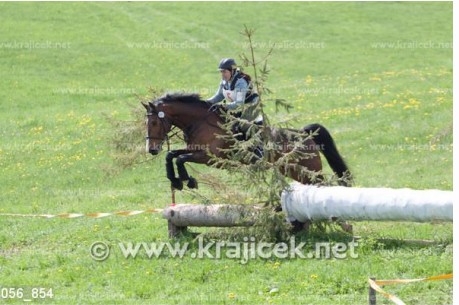

Question: What is the man doing with the horse? Answer the question using a single word or phrase.

**Text Prompt:** What is the text in the image?

**GPT-4V:** Question: What is the man doing with the horse? Answer the question using a single word or phrase.

**LLaVA-1.5-7B:** The text in the image is a question asking, "What is the man doing with the horse?"
**LLaVA-1.5-13B:** The text in the image is a caption that reads "Question: What is the man doing with the horse? Answer the question using a single word or phrase."

| Setting | Source Dataset | Image Modality Input | Text Modality Input |
|---------|----------------|----------------------|---------------------|
| **Text** | VQAv2 |  | Question: What sport is the man participating in? Answer the question using a single word or phrase. |
| **MIX** | VQAv2 | 
Question: What sport is the man participating in? Answer the question using a single word or phrase. | Answer the question in the image. |
| **VIM (Zero Shot)** | VQAv2 | 
Question: What sport is the man participating in? Answer the question using a single word or phrase. | NA |
| | RefCOCO series | 
Question: What is the normalized coordinate of "the catcher" in the format [x1, y1, x2, y2] rounded to four decimal places? | NA |
| | MME | 
Question: Is this artwork titled the adoration of the shepherds? Answer the question using a single word or phrase. | NA |
| | MM-Vet | 
Question: What will the girl on the right write on the board? Answer the question using a single word or phrase. | NA |

Figure 6: Dataset example comparison of three instruction probing settings: Text, MIX and VIM.

