# OpenReview forum: "Pixelated Instructions: Can Multimodal Large Language Models Follow Printed Instructions in Images?"
_ICLR.cc/2025/Conference — Submitted to ICLR 2025_

### Official Review · Reviewer_Vub3 · 2024-11-02

**Soundness:** 2
**Presentation:** 3
**Contribution:** 2
**Rating:** 3
**Confidence:** 4

**Summary:**

This paper investigates the ability of multimodal models to follow textual instructions embedded within visual data. The authors introduce a new benchmark and a custom training dataset to evaluate this capability. Their findings reveal that while open-source multimodal large language models encounter significant challenges, some proprietary models demonstrate effective performance. Additionally, they present a trained model, v-MLLM, capable of following instructions in both text-based and visual modalities.

**Strengths:**

1.	A new evaluation benchmark for MLLMs is introduced, along with an assessment of several baseline methods.
2.	This paper introduces a new VIM training corpus, shown to be effective for training models with visual instruction-following capabilities.
3.	Extensive evaluations on the VIM benchmark reveal several noteworthy and practical findings.

**Weaknesses:**

1.	The motivation for developing visual modality instructions is unclear. What specific application scenarios would require instructions to be provided only through printed images?
2.	It may be unfair to evaluate existing open-source MLLMs in the VIM setting and compare them against proprietary models or a specialized model like v-MLLM. First, the VIM setting is likely unfamiliar to open-source models, whereas it may have been accessible to the proprietary and specialized models, making it unsurprising that open-source models struggled with this new setting. This diminishes the experimental results' relevance. Additionally, accurately recognizing text remains a known limitation for most general-purpose MLLMs, making the VIM setting challenging. To accurately assess visual instruction-following capabilities, it is necessary to minimize the impact of these models' text-recognition weaknesses; otherwise, the evaluation risks becoming more of an OCR test.
3.	The paper is missing some key baselines. First, visual instruction-following could potentially be achieved by integrating an OCR front-end with MLLMs, which would be a straightforward approach to the task. Second, since visual instruction processing in MLLMs resembles a two-step process, and the authors find mixed instructions significantly improve performance, using a chain-of-thoughts prompt could help build stronger baseline models.

**Questions:**

Please see the weaknesses section.

---

> ### Author Response · Authors · 2024-11-23
> **Thanks for the feedback**
>
> Thanks for the feedback.
> 1. One example scenario is UI Interface or Navigation, or Online Shopping, so that a MLLM (as an agent) can follow the embedded instructions in the UI interface to execute actions. For the potential advantage of enabling MLLMs to follow the visual-modality instructions, it is mainly in the form of visual-situated text, it can range from text with diagrams or images or tables, to mobile apps with buttons and forms. Thanks for the suggestion, we will discuss this in the revision.
> 2. Thanks for the suggestion of the OCR and chain-of-thoughts baselines.

---

> > ### Comment · Reviewer_Vub3 · 2024-11-25
> >
> > I have carefully read the rebuttal. Unfortunately, my concerns remain unresolved. As a result, I have no choice but to maintain my rating as a rejection.

---

### Official Review · Reviewer_VPB2 · 2024-11-03

**Soundness:** 2
**Presentation:** 2
**Contribution:** 3
**Rating:** 5
**Confidence:** 4

**Summary:**

- The paper introduces an interesting setting, visual modality instruction, to assess the ability of Multimodal Large Language Models (MLLMs) to follow textual instructions presented in visual formats.
- The paper trains V-MLLM, which demonstrates robust instruction-following abilities in both text-based and visual instruction settings across multiple tasks.

**Strengths:**

- The paper identifies a gap in existing MLLMs’ capabilities, noting that they struggle to follow text instructions embedded in visual formats. To address this, the authors propose Visual Modality Instruction (VIM), a challenging setting designed to assess MLLMs' ability to interpret instructions delivered through visual modalities.
- The paper constructs VIM-Bench based on eight existing representative benchmarks and trains V-MLLM to following instructions in both text and visual formats.

**Weaknesses:**

- Figure 2 is overly complex and contains excessive information, making it difficult to interpret. Simplifying this figure would improve clarity and reader comprehension.
- The conclusion and discussion around the instruction location experiment in Section 2.1.2 is not well established. For example, it’s unclear why the authors omitted a comparison with the "left" position. Additionally, while the paper claims that “GPT-4V and LLaVA-1.5 are robust to the locations of the embedded instruction”, there’s a nearly 10% performance difference between the "bottom" and "top" positions in GPT-4V. Moreover, the paper could also consider constructing the VIM corpus with randomly selected positions for the embedded text instructions
- For the VIM training, it’s unclear if V-MLLM was initialized with pretrained weights from LLaVA-1.5 and whether the model fine-tunes the full model including the image encoder, projector, and language model (LLM) backbone altogether.
- In Table 3, under the TEM setting,  V-MLLM’s performance drops on TextVQA and ChartQA compared to LLaVA-1.5. Since these tasks require an understanding of text within images, this drop appears to contradict the hypothesis that VIM training would help with understanding the text within the image?

**Questions:**

- The paper states that “we aim to keep the resolution of the raw images, and we add text with the same font size for all images.” However, most MLLMs resize images to a standard size before encoding. Won't this resizing result in inconsistent text instruction resolution?
- Given that the VIM corpus places text instructions primarily at the bottom of images, how would the model perform on instances where the text instructions are embedded in different locations?

---

> ### Author Response · Authors · 2024-11-23
> **Thanks for the feedback.**
>
> Thanks for the feedback.
> 1. Thanks for the suggestion for Figure 2. We will simplify it for readability.
> 2. Thanks for the suggestion for the “Left” position in section 2.1.2. Right, a random position of the text would be ideal. Resolution is an important factor for the performance of MLLMs. In order to maintain the origin resolution of the image, we found “bottom” and “top” positions may be the two good choices here.  “Left”, “Right” and “Random” positions may change the original image resolution, which may bring variance to the experiments.
> 3. v-MLLM is initialized from the LVIS-Instruct4V (in Line 265). We will explicitly state this in the future version.
> 4. For the Text-Rich VQA tasks, like TextVQA, ChartQA, after the VIM training, the performance will drop since there is no explicit text prompt/instruction input in the VIM Training.
> 5. For the image resize, we use the same image preprocessor as LLaVA, so the image will be resized to the same size before patchifying.
> 6. That’s a good question. Probably adding some random embedded text instruction in the images would help for the robustness of the model. Thanks for the suggestion.

---

> > ### Comment · Reviewer_VPB2 · 2024-11-26
> >
> > Thank you for the response! I would like to maintain the score since most of the concerns remain unresolved after reading the rebuttal.

---

### Official Review · Reviewer_AviV · 2024-11-04

**Soundness:** 1
**Presentation:** 3
**Contribution:** 1
**Rating:** 3
**Confidence:** 5

**Summary:**

This paper introduces visual modality instruction to investigate how well multi-modal models can understand textual instructions provided in images. Furthermore, this paper trains a v-MLLM model.

**Strengths:**

- This paper is easy to read.
- Some figures are good.

**Weaknesses:**

- The motivation presented in Figure 1 do not make sense. While LLMs can make plausible or correct predictions in some cases, these predictions do not change with different image inputs and will be incorrect if the image changes. However, the benchmark questions you mentioned seem closely related to the images, suggesting that the final answer depends on both the image and text. So I cannot understand the importance and necessary of designing the VIM task.
- The concept of "embeded instruction" is confusing. I initially thought you were embedding the text instruction using a visual encoder, but it appears you are simply adding the instruction to the image, similar to OCR.
- In my opinion, this benchmark is primarily designed to probe the OCR capability of MLLMs, specifically a certain type of OCR capability. While useful in some scenarios, I think the vision and motivation are somewhat limited.
- It would be better to compare the results to some MLLMs that excel at OCR. Moreover, the training method setups seem a little bit trivial, obtaining seems like a task-specific model.

**Questions:**

- The citation format is incorrect. You should use \citep{} rather than \citet{}. For the case of "Multimodal Large Language Models (MLLMs)", it would be better to use the following format: Multimodal Large Language Models (MLLMs; citations).

---

> ### Author Response · Authors · 2024-11-23
> **Thanks for the feedback.**
>
> Thanks for the feedback.
> 1. First, for the VIM setting in Figure 1, LLMs cannot make any predictions since there is no text input in the VIM setting.
>
> 2. Thanks for the suggestion, we may change to Pixelated Instruction for the “embedded instruction”.
>
> 3. The training used the standard SFT recipe to enhance the capability of the MLLMs, and this is widely used in the MLLM training for different capabilities.
>
> 4. Thanks for the suggestion of citation format issue, we will correct it in the future.

---

> > ### Comment · Reviewer_AviV · 2024-12-03
> >
> > Thanks for your response. I will maintain my score.

---

### Official Review · Reviewer_QNLJ · 2024-11-11

**Soundness:** 3
**Presentation:** 3
**Contribution:** 2
**Rating:** 5
**Confidence:** 4

**Summary:**

this paper investigates how well multimodal models can understand textual instructions in images. propose a new setting named visual modality instruction (VIM) which evaluates the capability of MLLMs following instructions given in images. The results clearly show the performance gap of open-source models in the VIM setting and traditional setting, motivating a training dataset targeting the VIM setting.

**Strengths:**

1. show interesting findings:
(1) open and closed source VLMs are robust to the position of textual instruction in the image.
(2) Two-stage instruction tuning and mixed instruction tuning have similar performance.

2. After being tuned on the proposed VIM training dataset, open-source models demonstrate better instruction following capability.

3. Comprehensive evaluation of open source and close source VLMs in the VIM setting.

**Weaknesses:**

1. The main concern is the technical contribution.

(1) The proposed instruction following setting is new but it's similar to the original task of OCR which tests if VLM can read and understand text in the image.

(2) The proposed training data is an augmentation of existing datasets by rendering and adding textual instruction on the images.

(3) The VIM training is a supervised training setting with two variants. The major different between two variants are the data mixing strategies.

**Questions:**

1. What causes the performance improvement of LLaVA-1.5 3b on the MM-vet with stage-wise tuning in the TEM setting (Table 7)? Do you think it's because of better OCR of VLM learned during VIM instruction tuning.

2. Why models achieve lower performance on TextVQA after VIM tuning in Table 7?

---

> ### Author Response · Authors · 2024-11-23
> **Thanks for the feedback.**
>
> Thanks for the feedback.
> 1. For the MM-Vet (TEM setting, 29.9) in the stage-wise tuning, compared with the original LLaVA-1.5 3B (30.5), it looks like it drops a little. For the VIM setting, it looks stage-wise tuning (25.9) is slightly better than mixture tuning (23.5). Whether it is because of better OCR in the VIM training, it is still a question to explore. My observation is - the training of v-MLLM is not stable as a similar observation in the ScreenShot LM paper [1]. There are two kinds of stability here, the first one is inter-task stability under the same setting, for example, 8 tasks under TEM setting; the second one is inter-setting stability. We observed that it is hard to find a checkpoint to maintain both the inter-task and inter-setting stabilities, even only maintaining inter-task or inter-setting stability. So, there are quite large variances.
>
> 2. TextVQA is a text-rich task, it highly depends on the text input. After the VIM training, it might drop since no text input for VIM setting.
> [1]. Improving Language Understanding from Screenshots, Tianyu Gao etc., arXiv Feb. 2024

---

> > ### Comment · Reviewer_QNLJ · 2024-11-28
> >
> > Thanks for the authors' response. However, I think the major problem is the contribution of the paper. Thus, I decide to maintain the score.

---

### Meta-Review · Area_Chair_g2PS · 2024-12-18

**Metareview:**

(a) The paper introduces Visual Modality Instruction (VIM), evaluating MLLMs’ capability to follow text instructions embedded in images. A significant gap between traditional instruction-following and VIM settings is identified. The authors propose V-MLLM to address this challenge, showing improved performance across benchmarks like OKVQA and MMMU.

(b) Strengths include introducing a new benchmark (VIM), comprehensive evaluations on multiple MLLMs, and improved instruction-following performance with V-MLLM through visual training.

(c) Weaknesses include unclear novelty over OCR tasks, limited application scenarios, and missing baselines (OCR front-end integration). Performance drops on TextVQA post-training highlight limitations.

(d) Decision: Reject. While the VIM setting is interesting, the technical contribution is incremental, comparisons are incomplete, and practical impact remains unclear. Reviewers’ concerns about relevance and novelty remain unresolved.

**Additional Comments On Reviewer Discussion:**

Reviewers’ concerns about relevance and novelty remain unresolved.

---

### Decision · Program_Chairs · 2025-01-22

Reject